# Determinants of life satisfaction among Ghanaians aged 15 to 49 years: A further analysis of the 2017/2018 Multiple Cluster Indicator Survey

Kenneth Owusu Ansah[1], Nutifafa Eugene Yaw Dey[1]*, Abigail Esinam Adade[1], Pascal Agbadi[2,3]

1 Department of Psychology, University of Ghana, Legon-Accra, Ghana, 2 Department of Sociology and Social Policy, Lingnan University, Hong Kong, SAR, China, 3 Department of Nursing, Kwame Nkrumah University of Science and Technology, PMB, Kumasi, Ghana

◉ These authors contributed equally to this work.
* nutidey@gmail.com

**Data Availability Statement:** The data underlying the results presented in the study are available from MICS at https://mics.unicef.org/surveys.

## Abstract

The inclusion of life satisfaction in government policies as a tracker of the social and economic progress of citizens has been recommended. This has encouraged the scientific investigation of life satisfaction levels of people in tandem with factors responsible for these levels. Only a few studies have attempted to do this in Ghana with mixed findings. This study, therefore, extends previous literature by examining the determinants of life satisfaction among Ghanaians in two ways: a full sample and a gender-stratified sample. We analysed cross-sectional data from the 2017/2018 Ghana Multiple Indicator Cluster Survey Six (MICS 6). A sample of 20,059 women and men of ages ranging from 15 to 49 years participated in this study. The Cantril's Self-Anchoring Ladder Life Satisfaction scale was used to capture the life satisfaction of participants alongside relevant sociodemographic questions. About 35% of participants reported they were satisfied in life with males reporting more suffering levels [39.59%; 95% CI:36.38, 42.88] and females more thriving levels [36.41%; 95% CI:35.01, 37.84]. In the full sample multivariable model, gender, age, parity, education, marital status, wealth index, and region of residence were significantly associated with life satisfaction. Gender variations were also found across these associations. These findings collectively provide useful information for policymakers and practitioners to optimize interventions for the Ghanaian population aimed at improving life satisfaction. Evidence from this study also calls on the government of Ghana to begin tracking the life satisfaction of her citizens.

## Introduction

Life satisfaction refers to the subjective evaluation of life as a whole [1]. It implies that life satisfaction is a subjective cognitive appraisal of an individual's condition regarding several life

**Funding:** The author(s) received no specific funding for this work.

**Competing interests:** The authors have declared that no competing interests exist.

domains [2]. Researchers have expressed differing perspectives on the domains that comprise life satisfaction. Some scholars suggested that life satisfaction is dependent on a combination of both material and non-material factors such as one's health status, social relationship, family, and self-worth [3,4]. Others conceptualize life satisfaction based solely on material factors such as work, income, and place of residence [5,6].

Despite the divergent views about its domain, there seems to be a consensus in literature on the associated factors of life satisfaction from different parts of the world. In developed countries, factors such as household and personal income, health, age, gender, marital status, and education [5,7–12] have been identified as significant associates of life satisfaction. In Diego-Rosell, Tortora, and Bird's [9] study, for instance, it was found that families with a high household income had better life satisfaction. Luhmann et al. [11] also found from analysing longitudinal data from three nationally representative panel studies that life satisfaction levels increased with marriage and childbirth but reduced with marital separation, job loss, starting a new job, and relocating.

Similar findings on associated factors have been reported in studies from Sub-Saharan Africa, particularly South Africa, Malawi, Ethiopia, Nigeria, and Ghana [13–17]. In the Ghanaian context, the few existing evidence shows that being involved in religion, experiences of migration, having a high income, higher education, social capital, tobacco use, being in the lower class, residing in the southern parts, job security and being married were related to life satisfaction [13,18–26].

While these few Ghanaian studies provide preliminary evidence, they also suggest the need for more investigation into the factors associated with life satisfaction. Moreover, the majority of these studies have focused on only one aspect or domain of life satisfaction. For instance, Pokimica et al. [21], as well as Addai and Pokimica [18], operationalized life satisfaction in terms of the living standards of individuals rather than how individuals feel about their overall life. So far, only Addai et al. [13] and Calys-Tagoe et al. [20], the most recent studies in Ghana, have measured life satisfaction as the extent to which individuals feel about their overall life. Even though both studies relied on a nationally representative sample of men and women, there are some methodological limitations worth mentioning. The age of Addai et al's [18] dataset is now quite old, using data that was collected within 2005–2008. Also, Calys-Tagoe et al' study [20] only focused on older adults (50 years and above).

Considering these limitations and the pressing need to produce more recent evidence in particular because of persisting contextual problems ranging from limited access to drinking water [27], unemployment particularly among the youth [28], limited access to health care, poverty [29], high prevalence of chronic diseases [30–34] and poor quality education [35,36] that may threaten one's life satisfaction, our study used the 2017/2018 Multiple Cluster Indicator Survey to examine the factors associated with life satisfaction in Ghana. Our study goes a step further by examining closely these factors from a gendered perspective; an examination that is virtually non-existent in Ghana. Due to the variations in social norms and biological characteristics, a gendered perspective has been recommended to provide more nuanced information into the associated factors of both men and women's life satisfaction [37–39]. The outcome of such attempt has been highlighted by Joshanloo [37] who concluded that sociopolitical, employment-related, and education-related variables were more important to men's life satisfaction whereas women's life satisfaction was associated with variables related to marital status and interpersonal relationships. It is, therefore, reasonable to expect that such unique differences may exist in our study. The findings of this study will be useful for policymakers, researchers, and practitioners in designing gender-specific interventions and services to improve the life satisfaction of men and women in Ghana.

## Methods

### Study design and data source

This study used existing data from the 2017–2018 Ghana Multiple Indicator Cluster Survey Six (MICS 6). The Ghana MICS is a cross-sectional survey conducted by the Ghana Statistical Service (GSS) in collaboration with the Ghana Health Service (GHS), Ministry of Health (MOH), and the Ministry of Education with funding and technical support being provided by UNICEF and other international donors [40]. In the 1990s, UNICEF launched the Global MICS Programme as an international multi-purpose household survey initiative to assist countries in gathering internationally comparable data on a wide range of initiatives on the situation of children and women. MICS analyses key indicators that assist countries to generate data for use in national development plans, policies, and programmes, as well as to measure progress towards SDGs and other agreements signed internationally [40].

A multi-stage, stratified cluster sampling approach was used to nationally survey children and women in urban and rural areas, located in the previously 10 administrative regions in Ghana: Western, Central, Greater Accra, Volta, Eastern, Ashanti, Brong Ahafo, Northern, Upper East and Upper West [40]. The sampling frame for data collection was based on the 2010 Population and Housing Census (2010 PHC) of Ghana. At the first stage guided by the definition of the 2010 PHC enumeration, the enumeration areas (EAs) were identified within the selected primary sampling units (PSUs). In each EA sample, the cataloguing of households was carried out and a sample of households was selected in the second stage using systematic random sampling. In each sampled household, all persons who met the inclusion criteria (e.g., age 15–49 years) were eligible to participate in the survey. The smallest regions were allocated a minimum of 60 sample clusters (primary sampling units), and the Greater Accra Region was allocated a maximum of 86 sample clusters. The sample clusters were distributed between the urban and rural strata within each area, proportionate to the size of the corresponding populations within the frame. Clusters (primary sampling units) were assigned to the urban and rural strata in each area in proportion to the number of households in the census frame for each stratum within that region [40]. The final samples were 660 clusters and 13202 households across all sampling strata.

### Ethics

UNICEF through the Ghana Statistical Service obtained ethical clearance. Verbal consent and assent were obtained from participants age 18+ and 15-17years, respectively. All participants were informed about the voluntary nature of participation including confidentiality and anonymity.

### Study sample

Our full sample size totalled 20,059 women and men of ages ranging from 15 to 49years. Women and men numbered 14,587 and 5,472, respectively.

### Measures

On behalf of the GSS and UNICEF, trained enumeration officials collected the data. Six questionnaires were included in the field data collection instrument: 1) Household questionnaire, 2) Water Quality Testing Questionnaire, 3) Questionnaire for Individual Women, 4) Questionnaire for Individual Men, 5) Questionnaire for Children Under Five, and 6) Questionnaire for Children Age 5–17. We used data collected with Individual Women and Individual Men questionnaires that were administered to randomly selected women and men living in the surveyed household.

**Outcome variable.**   The outcome variable, life satisfaction, was measured using the Cantril's Self-Anchoring Ladder of Life Satisfaction scale in the MICS. An image of a ladder with steps numbered from '0' at the bottom to '10' at the top was shown to participants and they were asked to indicate at which step of the ladder they believed they stood about their level of life satisfaction at the time of the survey. Following the recommendation of Cantril [41] and Gallup [42], the responses were categorized as suffering (0–4) as '0', struggling (5–6) as '1', and thriving (7–10) as '2'. This recategorized variable was kept solely for the purposes of description; the original variable (i.e., the 0 to 10 ordinal measurement) was used in the main study analyses.

**Predictor variables.**   The following variables were selected as predictor variables: gender, age, marital status, education, happiness level, health insurance, rural-urban, household wealth, and region of residence. Our selection was based on reports from prior research and variable availability [13,20]. We maintained the original categorization of each of these variables from the dataset. See Table for the categorization. Detailed explanations for these variables are provided elsewhere [40]. Briefly, most of the selected variables were measured in a simple manner (e.g., "Are you covered by any health insurance?" with responses "Yes" or "No") while others were aggregated from responses to several questions like the computation of household wealth of participants using household characteristics, possessions and assets (e.g., internet access, number of rooms for sleeping, source of drinking water, ownership of television, radio, vehicles and access to electricity, among others). Household wealth was categorized into poorest (0), second quintile (1), middle (2), fourth quintile (3), and richest (4).

## Data preparation and analysis

Data analysis began by recoding the selected variables after appending both male and female datasets in Stata version 14. Spearman's rho was used as a preliminary test to assess the inter-correlations between all study variables (see S1 Table). Next, we weighted the data to perform univariable analysis computing for frequency distributions of study variables. We accounted for the complex sampling design embedded in the dataset to account for possible analytical errors and make proper inferences [43]. This was achieved by correcting for clusters, stratification, and sample weights using the complex survey mode command 'svyset'. After this correction, we performed bivariable (see S2 Table) and multivariable analyses, regressing life satisfaction (as it was originally measured with the Cantril's ladder) onto the predictor variables on the full and gender-stratified samples using ordered probit regression ('oprobit') command. The ordered probit model is typically used to examine the variation in the data points of an ordinal categorical dependent variable [44]. Though argued to produce parameter estimates difficult to interpret, oprobit was fitted mainly for its ability to preserve the ordering of the response options in the outcome variable as a function of the predictor variables [45]. Next, we ran the margins command to produce predicted probabilities (see Table 3) only for the gender-stratified models and to ease interpretation of the estimated coefficient from the oprobit output [44]. Additionally, margins plots were generated for the highest level of life satisfaction to further support the interpretation of the predicted probabilities. We decided to make the plots more compact by interacting gender with selected predictors including age groups, education level, household wealth index and marital status using the full sample model (see S1 Fig). We report only adjusted models, pegging statistical significance at p≤0.05.

## Results

The descriptive results showed that about 35% [95% CI:33.49, 35.95] of all respondents are in the thriving category of the Cantril's Self-Anchoring Ladder of Life Satisfaction scale. Nonetheless, there are some gender differences. As shown in Fig 1, about 40% [95% CI:36.38, 42.88] of

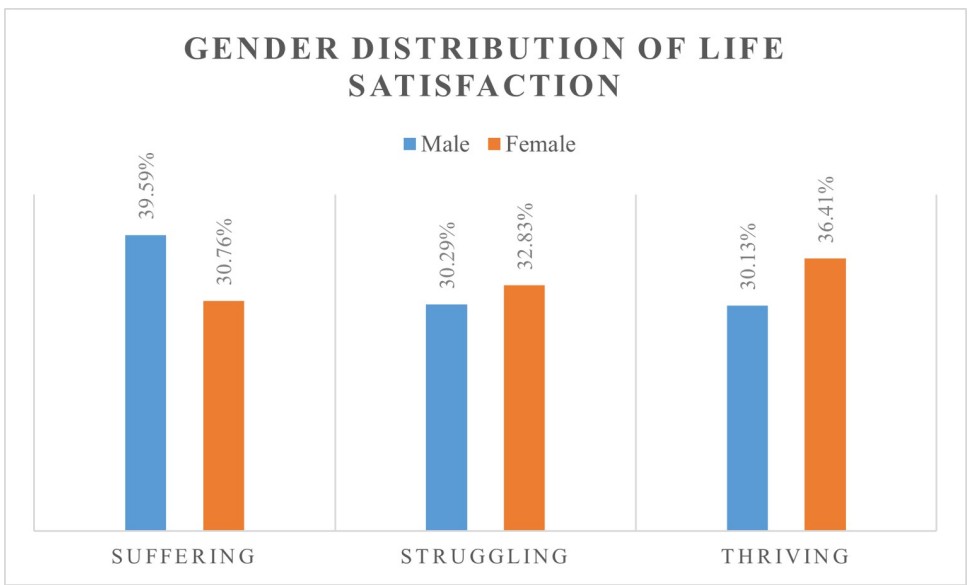

**Fig 1. Gender distribution of life satisfaction in Ghana.**

males and 31% [95% CI:29.3, 32.25] of females reported that they were suffering while 30% [95% CI:27.85, 32.5] of males and 36% [95% CI:35.01, 37.84] of females reported that they were thriving. This interestingly suggests that more males are suffering than females and on the contrary, more females than males are thriving. The majority of respondents have obtained a middle school level education (8044/19670). About 51% have health insurance (10136/19670). Detailed information on the study variables is presented in Table 1.

## Associated factors of life satisfaction in a gender-stratified and full sample multivariable models

In the full sample multivariable model, gender, age, education, marital status, parity, wealth index, and region of residence were significantly associated with life satisfaction (see Table 2). Differences existed in the determinants of life satisfaction in the gender-stratified sample models. The association between age and life satisfaction was relatively the same for both males and females; the results revealed that males age group of 25–29 and females within the age groups of 20–24, 25–29, and 40–44 years had a reduced probability of reporting higher life satisfaction compared with those within the age group of 45–49 years. Education was also associated with life satisfaction. The findings revealed that having a higher education status increases males' and females' probability of reporting higher life satisfaction levels. The association between marital status, parity, household wealth index, region of residence, and life satisfaction was different for females and males. For instance, the probability of reporting higher life satisfaction was higher for women who were currently married compared with previously married women. For men, both those who were currently married/in union and never married/in union were associated with an increase in the probability of reporting higher life satisfaction levels. Regarding parity, women who had one child and more had a reduced probability of being satisfied with life compared with women with no children. However, parity was not statistically significantly related to life satisfaction among males. Higher life satisfaction levels were also reported by males who were covered by health insurance than males without insurance, although this association was not significant for females. Males belonging to the richest wealth quintile while females belonging to both the fourth

**Table 1. Summary statistics of study variables.**

|  | Full sample | Male sample | Female sample |
|---|---|---|---|
| **Variables** | *n* (%) | *n* (%) | *n* (%) |
| **Life satisfaction** |  |  |  |
| [0] 0–4: Suffering | 6520 (33.14) | 2106 (39.59) | 4414 (30.76) |
| [1] 5–6: Struggling | 6323 (32.15) | 1611 (30.29) | 4712 (32.83) |
| [2] 7–10: Thriving | 6828 (34.71) | 1603 (30.13) | 5225 (36.41) |
| **Gender** |  |  |  |
| [0] Male | 5323 (27.0) |  |  |
| [1] Female | 14374 (73.0) |  |  |
| **Age** |  |  |  |
| [0] 15–19 years | 4413 (22.4) | 1487 (27.93) | 2926 (20.36) |
| [1] 20–24 years | 3106 (15.8) | 911 (17.2) | 2195 (15.27) |
| [2] 25–29 years | 2724 (13.8) | 568 (10.68) | 2156 (15.00) |
| [3] 30–34 years | 2795 (14.2) | 647 (12.16) | 2148 (14.94) |
| [4] 35–39 years | 2550 (12.9) | 617 (11.59) | 1933 (13.45) |
| [5] 40–44 years | 2256 (11.5) | 557 (10.46) | 1699 (11.82) |
| [6] 45–49 years | 1852 (9.4) | 535 (10.05) | 1317 (9.16) |
| **Education** |  |  |  |
| [0] Pre-primary or none | 3228 (16.4) | 525 (9.86) | 2703 (18.81) |
| [1] Primary | 3141 (15.9) | 633 (11.89) | 2508 (17.45) |
| [2] Junior Secondary | 8044 (40.8) | 2280 (42.84) | 5764 (40.10) |
| [3] Senior Secondary | 3948 (20.0) | 1382 (25.95) | 2566 (17.86) |
| [4] Higher | 1335 (6.8) | 504 (9.46) | 831 (5.78) |
| **Marital status** |  |  |  |
| [1] Never married/in union | 7526 (38.2) | 2724 (51.17) | 4802 (33.41) |
| (2) Currently married/in union | 10606 (53.8) | 2402 (45.12) | 8204 (57.08) |
| [0] Formerly married/in union | 1565 (7.9) | 198 (3.72) | 1367 (9.51) |
| **Parity** |  |  |  |
| [0] No child | 7225 (36.68) | 2818 (52.94) | 4407 (30.66) |
| [1] One child | 2248 (11.41) | 419 (7.88) | 1829 (12.72) |
| [2] Two children | 2203 (11.18) | 443 (8.33) | 1760 (12.24) |
| [3] Three children | 2094 (10.63) | 419 (7.87) | 1675 (11.65) |
| [4] Four + children | 5927 (30.09) | 1223 (22.98) | 4704 (32.72) |
| **Health Insurance coverage** |  |  |  |
| [0] Without insurance | 9561 (48.5) | 3182 (59.78) | 6379 (44.38) |
| [1] With insurance | 10136 (51.5) | 2141 (40.22) | 7995 (55.62) |
| **Household wealth index** |  |  |  |
| [0] Poorest | 3370 (17.1) | 969 (18.21) | 2401 (16.70) |
| [1] Second | 3534 (17.9) | 870 (1.34) | 2664 (18.54) |
| [2] Middle | 4020 (20.4) | 1106 (20.78) | 2914 (20.27) |
| [3] Fourth | 4243 (21.5) | 1202 (22.59) | 3041 (21.16) |
| [4] Richest | 4529 (23.0) | 1176 (22.08) | 3353 (23.33) |
| **Rural-Urban residence** |  |  |  |
| [0] Rural | 9896 (50.2) | 2811 (52.81) | 7085 (49.29) |
| [1] Urban | 9801 (49.8) | 2512 (47.19) | 7289 (50.71) |
| **Region of residence** |  |  |  |
| [0] Western | 1940 (9.8) | 520 (9.77) | 1420 (9.88) |
| [1] Central | 1867 (9.5) | 459 (8.63) | 1407 (9.79) |

(*Continued*)

**Table 1.** (Continued)

| | Full sample | Male sample | Female sample |
|---|---|---|---|
| **Variables** | **n (%)** | **n (%)** | **n (%)** |
| [2] Greater Accra | 2531 (12.8) | 642 (12.06) | 1889 (13.14) |
| [3] Volta | 1531 (7.8) | 426 (8.01) | 1105 (7.68) |
| [4] Eastern | 2401 (12.2) | 680 (12.77) | 1721 (11.97) |
| [5] Ashanti | 4744 (24.1) | 1305 (24.52) | 3439 (23.93) |
| [6] Brong Ahafo | 1787 (9.1) | 472 (8.87) | 1315 (9.15) |
| [7] Northern | 1839 (9.3) | 517 (9.72) | 1322 (9.20) |
| [8] Upper East | 590 (3.0) | 164 (3.08) | 426 (2.96) |
| [9] Upper West | 467 (2.4) | 137 (2.57) | 330 (2.30) |

and richest wealth quintile had an increased probability of reporting higher life satisfaction levels compared with those in the poorest wealth quintile.

There were also variations in the association between region of residence and life satisfaction among males and females. First, in both the male and the female samples, residing in Volta and Greater Accra regions were associated with a higher probability of life satisfaction compared with the reference region (Western). The associated differences between region of residence and life satisfaction among males and females are as follows. Compared to women residents of the Western region, women residents of Brong Ahafo, Eastern, Northern, Upper East, and Upper West regions had an increased probability of reporting higher life satisfaction levels. However, the relationship between residing in these regions on life satisfaction was not significant for males. Similarly, compared to men residents of the Western region, men residents of the Central region had a reduced probability of reporting higher life satisfaction. However, this relationship was not significant for females.

## Predicted probabilities of associated factors of life satisfaction

For better interpretation, Table 3 supplements the multivariable oprobit results by reporting the average marginal effect of the predictor variables [44]. Because the dependent variable takes all levels of life satisfaction into account, the estimations of the marginal effect yielded 11 sets of results. However, only three points on the life satisfaction ladder namely lowest, middle and highest are presented to save space. The predicted probabilities were interpreted by comparing the probabilities of a variable's reference category with the probabilities of other categories. From Table 3, the predicted probability of reporting higher satisfaction levels for males aged 15–19, 20–24, 25–29, 30–34, 35–39 and 40–44 years is 0.071, 0.051, 0.048, 0.060, 0.061 and 0.068, respectively, which were lower than the 0.074 of males aged 45-49years, holding all other variables constant. Similarly, the probability of reporting the highest life satisfaction for women aged 15–19, 20–24, 25–29, 30–34, 35–39 and 40–44 years was 0.121, 0.102, 0.101, 0.121, 0.115 and 0.102, respectively, lower than the 0.131 of women aged 45-49years. Lastly, compared with those without formal education or only earned pre-primary education, we find that having higher education on average increases the probability of identifying as highly satisfied with life by 0.09 for males and 0.17 for females. This finding conversely mirrors life satisfaction at its lowest level, where highly educated males (0.011) and females (0.014) had reduced average probabilities of reporting lowest life satisfaction compared to males (0.022) and females (0.028) with no or pre-primary education, holding all other variables constant. Full details of the other variables are reported in Table 3. The predicted probabilities for age groups, education level, wealth quintile and marital status at the highest levels of life satisfaction are plotted in S1 Fig.

**Table 2. Multivariate oprobit model regressing life satisfaction on predictor variables in the full sample and gender stratified sample.**

| | Full Sample | Male sample | Female sample |
|---|---|---|---|
| **Predictors** | Coef. [95% CI] | Coef. [95% CI] | Coef. [95% CI] |
| **Gender** | | | |
| Male | [ref] | NA | NA |
| Female | 0.23*** [0.16, 0.29] | | |
| **Age** | | | |
| 15–19 years | -0.03 [-0.14, 0.081] | -0.02 [-0.25, 0.21] | -0.05 [-0.18, 0.08] |
| 20–24 years | -0.16** [-0.26, -0.06] | -0.19 [-0.39, 0.004] | -0.16** [-0.27, -0.04] |
| 25–29 years | -0.18*** [-0.27, -0.09] | -0.23* [-0.41, -0.04] | -0.16** [-0.26, -0.05] |
| 30–34 years | -0.07 [-0.17, 0.03] | -0.11 [-0.26, 0.04] | -0.05 [-0.17, 0.07] |
| 35–39 years | -0.10* [-0.18, -0.017] | -0.11 [-0.30, 0.08] | -0.08 [-0.17, 0.01] |
| 40–44 years | -0.14** [-0.24, -0.035] | -0.05 [-0.21, 0.11] | -0.15* [-0.27, -0.04] |
| 45–49 years | [ref] | [ref] | [ref] |
| **Education** | | | |
| Pre-primary or none | [ref] | [ref] | [ref] |
| Primary | 0.03 [-0.06, 0.13] | 0.01 [-0.22, 0.24] | 0.04 [-0.10, 0.17] |
| Junior Secondary | 0.03 [-0.05, 0.11] | 0.07 [-0.14, 0.27] | 0.02 [-0.09, 0.13] |
| Senior Secondary | 0.02 [-0.07, 0.11] | -0.03 [-0.24, 0.18] | 0.05 [-0.07, 0.16] |
| Higher | 0.28*** [0.17, 0.39] | 0.27* [0.038, 0.51] | 0.29*** [0.14, 0.44] |
| **Marital Status** | | | |
| Currently married/in union | 0.26*** [0.18, 0.34] | 0.41* [0.062, 0.76] | 0.24*** [0.17, 0.32] |
| Formerly married/in union | [ref] | [ref] | [ref] |
| Never married/in union | 0.12* [0.001, 0.23] | 0.42* [0.04, 0.81] | 0.05 [-0.06, 0.17] |
| **Parity** | | | |
| No child | [ref] | [ref] | [ref] |
| One child | -0.09* [-0.17, -0.004] | -0.06 [-0.27, 0.15] | -0.09* [-0.18, -0.001] |
| Two children | -0.14** [-0.23, -0.04] | -0.04 [-0.25, 0.18] | -0.17** [-0.28, -0.06] |
| Three children | -0.10* [-0.20, -0.01] | 0.02 [-0.20, 0.24] | -0.15* [-0.26, -0.03] |
| Four + children | -0.08 [-0.17, 0.02] | 0.02 [-0.17, 0.22] | -0.12* [-0.23, -0.01] |
| **Insurance coverage** | | | |
| Without insurance | [ref] | [ref] | [ref] |
| With insurance | 0.06** [0.02, 0.11] | 0.16*** [0.08, 0.25] | 0.02 [-0.03, 0.07] |
| **Household wealth** | | | |
| Poorest | [ref] | [ref] | [ref] |
| Second | -0.05 [-0.14, 0.04] | -0.13 [-0.27, 0.01] | -0.02 [-0.13, 0.09] |
| Middle | 0.02 [-0.07, 0.11] | -0.13 [-0.27, 0.02] | 0.07 [-0.04, 0.18] |
| Fourth | 0.12* [0.03, 0.20] | -0.004 [-0.15, 0.15] | 0.15** [0.04, 0.26] |
| Richest | 0.29*** [0.20, 0.39] | 0.28** [0.11, 0.45] | 0.30*** [0.19, 0.41] |
| **Rural-Urban** | | | |
| Rural | [ref] | [ref] | [ref] |
| Urban | -0.003 [-0.07, 0.06] | -0.07 [-0.22, 0.08] | 0.02 [-0.06, 0.10] |
| **Region of residence** | 0 | 0 | 0 |
| Western | [ref] | [ref] | [ref] |
| Central | -0.01 [-0.13, 0.11] | -0.32* [-0.58, -0.05] | 0.08 [-0.05, 0.21] |
| Greater Accra | 0.21*** [0.11, 0.32] | 0.27* [0.04, 0.51] | 0.19** [0.06, 0.31] |
| Volta | 0.31*** [0.16, 0.46] | 0.29* [0.047, 0.54] | 0.33*** [0.15, 0.50] |
| Eastern | 0.12* [0.01, 0.23] | -0.09 [-0.31, 0.14] | 0.19** [0.06, 0.33] |

(*Continued*)

**Table 2.** (Continued)

| Predictors | Full Sample | Male sample | Female sample |
|---|---|---|---|
| | Coef. [95% CI] | Coef. [95% CI] | Coef. [95% CI] |
| Ashanti | 0.03 [-0.09, 0.14] | -0.04 [-0.29, 0.21] | 0.05 [-0.08, 0.17] |
| Brong Ahafo | 0.22*** [0.11,0.34] | 0.08 [-0.17, 0.32] | 0.28*** [0.15, 0.41] |
| Northern | 0.35*** [0.22, 0.49] | 0.25 [-0.02, 0.53] | 0.38*** [0.23, 0.52] |
| Upper East | 0.51*** [0.37, 0.65] | 0.06 [-0.23, 0.35] | 0.67*** [0.52, 0.83] |
| Upper West | 0.40*** [0.26, 0.55] | 0.33 [-0.02, 0.68] | 0.42*** [0.28, 0.56] |
| $F$ statistics | $F(32, 609) = 17.91$, $p<0.001$ | $F(31, 610) = 8.36$, $p<0.001$ | $F(31, 610) = 11.79$, $p<0.001$ |

*Note.*

* $p < 0.05$

** $p < 0.01$

*** $p < 0.001$

95% CI: Confidence interval; Coef.: Robust regression coefficient.

## Discussion

This study examined determinants of life satisfaction among Ghanaians aged 15 to 49 years. In the full sample multivariable model, gender, age, education, marital status, parity, wealth index, and region of residence were significantly associated with life satisfaction. Same relationships existed in the gender-stratified samples. However, there were some slight variations across genders. This will be the focus of our discussion.

Within the age groups, those who were aged 20–24, 25–29, and 40–44 years had a reduced probability to be satisfied with life compared with those within the age group of 45–49 years. We found that the results were similar for both young male and female adults. However, only older females aged 40–44 reported lower levels of life satisfaction. This age pattern (which is depicted in S1 Fig) clearly mirrors the U-shape well-being curve indicating that happiness declines from late adolescence and rises in midlife [7,46]. Transitioning from adolescence to early adulthood is a vulnerable period in which young people take their first tentative steps toward independence. This phase (20–29 years) is often associated with significant life changes and responsibilities as an individual works toward his or her goals including emancipation, getting married, getting a higher education, and employment [47]. Actualizing these goals tends to put pressure and stress on young individuals and severely impairs their well-being and life satisfaction [47,48]. In responding to life stressors, some young adults may engage in risky health behaviours such as substance use [49]. Frequent usage of substances such as smoking cigarettes, marijuana, or drinking alcohol which is commonest among young adults than older adults [50], increases the likelihood of low life satisfaction [51]. For females between the ages of 40 and 44 fertility declines and the onset of menopausal syndrome begins [52]. The desire to have children, as well as initial responses to these menopausal symptoms such as depression, hot flashes, and insomnia, may explain their decreases in life satisfaction [53,54].

Being highly educated was related to perceived higher life satisfaction for both men and women. This finding agrees with Powdthavee et al. [55] who reported higher levels of life satisfaction among the highly educated. According to Maslow's hierarchy of needs [56] after the satisfaction of basic needs such as food, water, shelter, and clothing, the next higher human needs among others include higher forms of education. It is believed that the gratification of this need comes with higher levels of life satisfaction [56,57]. For instance, education serves as a springboard for better career opportunities and reduces the risk of being unemployed [58].

**Table 3. Predicted probabilities for male and female sample at 3 points (lowest, middle and highest) of life satisfaction ladder.**

| Predictors | Male (Margins [Std. E]) | | | Female (Margins [Std. E]) | | |
|---|---|---|---|---|---|---|
| | Lowest (0) | Middle (5) | Highest (10) | Lowest (0) | Middle (5) | Highest (10) |
| **Age** | | | | | | |
| 15–19 years | .017*** (.003) | .195*** (.010) | .071*** (.010) | .023*** (.003) | .213*** (.006) | .121*** (.009) |
| 20–24 years | .025*** (.003) | .195*** (.010) | .051*** (.006) | .029*** (.003) | .217*** (.006) | .102*** (.007) |
| 25–29 years | .027*** (.004) | .194*** (.010) | .048*** (.007) | .029*** (.003) | .217*** (.006) | .101*** (.006) |
| 30–34 years | .020*** (.003) | .195*** (.010) | .060*** (.008) | .023*** (.003) | .213*** (.007) | .121*** (.013) |
| 35–39 years | .020*** (.005) | .195*** (.010) | .061*** (.009) | .025*** (.003) | .214*** (.006) | .115*** (.007) |
| 40–44 years | .017*** (.004) | .195*** (.010) | .068*** (.009) | .030*** (.005) | .217*** (.006) | .102*** (.009) |
| 45–49 years | .015*** (.004) | .194*** (.010) | .074*** (.010) | .020*** (.003) | .210*** (.006) | .131*** (.009) |
| **Education** | | | | | | |
| Pre-primary or none | .022*** (.006) | .196*** (.010) | .056*** (.011) | .028*** (.003) | .217*** (.007) | .105*** (.010) |
| Primary | .021*** (.003) | .196*** (.010) | .057*** (.007) | .025*** (.003) | .215*** (.006) | .111*** (.007) |
| Junior Secondary | .019*** (.003) | .196*** (.010) | .064*** (.006) | .026*** (.003) | .216*** (.006) | .108*** (.005) |
| Senior Secondary | .023*** (.004) | .195*** (.010) | .053*** (.005) | .025*** (.003) | .215*** (.006) | .113*** (.007) |
| Higher | .011*** (.002) | .191*** (.010) | .092*** (.011) | .014*** (.002) | .198*** (.007) | .165*** (.011) |
| **Marital Status** | | | | | | |
| Currently married/in union | .019*** (.003) | .196*** (.010) | .063*** (.007) | .021*** (.002) | .210*** (.006) | .130*** (.006) |
| Formerly married/in union | .047** (.019) | .183*** (.019) | .027** (.011) | .036*** (.004) | .220*** (.006) | .086*** (.007) |
| Never married/in union | .019*** (.003) | .196*** (.010) | .064*** (.007) | .031*** (.003) | .219*** (.006) | .094*** (.007) |
| **Parity** | | | | | | |
| No child | .020*** (.003) | .195*** (.010) | .063*** (.007) | .022*** (.002) | .210*** (.006) | .130*** (.008) |
| One child | .023*** (.005) | .195*** (.010) | .056*** (.010) | .025*** (.003) | .215*** (.006) | .113*** (.007) |
| Two children | .022*** (.004) | .195*** (.010) | .058*** (.010) | .030*** (.004) | .217*** (.006) | .098*** (.007) |
| Three children | .019*** (.004) | .195*** (.010) | .065*** (.011) | .029*** (.003) | .217*** (.006) | .102*** (.008) |
| Four children | .018*** (.004) | .195*** (.010) | .065*** (.009) | .027*** (.003) | .216*** (.006) | .107*** (.007) |
| **Insurance coverage** | | | | | | |
| Without insurance | .022*** (.003) | .196*** (.010) | .054*** (.005) | .026*** (.002) | .215*** (.006) | .111*** (.006) |
| With insurance | .015*** (.002) | .195*** (.010) | .073*** (.006) | .025*** (.002) | .214*** (.006) | .114*** (.005) |
| **Household wealth index** | | | | | | |
| Poorest | .019*** (.003) | .198*** (.010) | .058*** (.007) | .032*** (.004) | .220*** (.006) | .092*** (.008) |
| Second | .026*** (.005) | .196*** (.011) | .045*** (.006) | .033*** (.004) | .220*** (.006) | .089*** (.007) |
| Middle | .026*** (.004) | .196*** (.010) | .045*** (.006) | .027*** (.003) | .218*** (.006) | .104*** (.008) |
| Fourth | .019*** (.003) | .198*** (.010) | .058*** (.007) | .023*** (.003) | .214*** (.006) | .119*** (.007) |
| Richest | .010*** (.002) | .190*** (.010) | .098*** (.010) | .016*** (.002) | .204*** (.006) | .150*** (.008) |
| **Rural-Urban** | | | | | | |
| Rural | .019*** (.002) | .194*** (.010) | .067*** (.007) | .026*** (.003) | .215*** (.006) | .111*** (.007) |
| Urban | .022*** (.004) | .194*** (.010) | .059*** (.006) | .025*** (.003) | .214*** (.006) | .114*** (.006) |
| **Region of residence** | | | | | | |
| Western | .021*** (.005) | .198*** (.010) | .054*** (.011) | .037*** (.005) | .221*** (.006) | .081*** (.008) |
| Central | .041*** (.007) | .187*** (.011) | .028*** (.006) | .031*** (.004) | .219*** (.006) | .094*** (.008) |
| Greater Accra | .011*** (.002) | .193*** (.010) | .090*** (.012) | .024*** (.003) | .216*** (.006) | .112*** (.008) |
| Volta | .010*** (.002) | .192*** (.010) | .093*** (.012) | .018*** (.004) | .207*** (.007) | .141*** (.016) |
| Eastern | .025*** (.004) | .197*** (.010) | .046*** (.006) | .024*** (.003) | .215*** (.007) | .113*** (.010) |
| Ashanti | .022*** (.005) | .198*** (.011) | .050*** (.008) | .034*** (.004) | .220*** (.006) | .088*** (.007) |
| Brong Ahafo | .017*** (.004) | .198*** (.010) | .062*** (.010) | .020*** (.003) | .211*** (.007) | .130*** (.010) |
| Northern | .011*** (.003) | .194*** (.011) | .087*** (.014) | .016*** (.003) | .203*** (.007) | .152*** (.012) |
| Upper East | .018*** (.005) | .198*** (.010) | .060*** (.013) | .007*** (.001) | .173*** (.008) | .231*** (.018) |

*(Continued)*

**Table 3.** (Continued)

|  | Male (Margins [Std. E]) | | | Female (Margins [Std. E]) | | |
| --- | --- | --- | --- | --- | --- | --- |
| Predictors | Lowest (0) | Middle (5) | Highest (10) | Lowest (0) | Middle (5) | Highest (10) |
| Upper West | .009** (.004) | .190*** (.012) | .099*** (.024) | .014*** (.002) | .200*** (.007) | .162*** (.012) |

*Note.*

* $p < 0.05$

** $p < 0.01$

*** $p < 0.001$

Robust standard errors are in parentheses.

Job opportunities and higher incomes serve as indirect conduits through which education increases life satisfaction [57,59]. On the other hand, life dissatisfaction, anger, frustration, and unhappiness have been associated with unemployment [60]. This finding is consistent with previous research indicating that higher education is significantly related to the degree to which both men and women are satisfied with their lives [61].

We also found that married men and women were more satisfied with life than those who were previously married. While this contradicts the findings of Addai et al. [19], who found that marriage is negatively associated with happiness and life satisfaction in the Ghanaian context, it supports the findings of Botha and Booysen's [62] study, which reports that married people in South Africa have higher levels of life satisfaction than divorced people. Marriage brings about love, mutual support, protection, social control, economic benefits, and all these tend to increase life satisfaction [63]. Marriage in Ghana is a social event guided by customary law and held in high esteem [19,64]. This is not surprising because a study by Dery and Bawa revealed that women in Northern Ghana were more satisfied with life mostly because marriage is revered by women and is associated with some form of prestige [65]. Furthermore, in Ghana, men are expected to marry at a certain age (young-middle adulthood) to be considered responsible and "men" [65,66]. Once this is done, some level of prestige is established, and societal pressure reduces. The study's findings also revealed that unmarried men equally reported higher levels of satisfaction. The single status also has advantages: it is associated with fewer responsibilities, greater individual growth, and independence, greater freedom, more energy towards career goals, more time for friendship, fewer marital issues such as domestic fighting, and greater peace [67,68] and these contribute to an increase in life satisfaction.

Females with one and more children reported reduced life satisfaction compared to their counterparts who had no children. This finding is consistent with the findings showing that mothers having and raising children may experience reduced life satisfaction [69,70]. This is because children demand lots of attention and time. Having children involves providing care and performing household duties such as shopping, washing, cleaning, and cooking. This increased workload is especially true for Ghanaian women because women are not only responsible for catering for their children but work outside their homes in a variety of formal and informal occupations [71]. The increased demands of juggling between child-rearing and careers may account for the negative impact on life satisfaction among females with one or more children.

Furthermore, our findings revealed that men with health insurance coverage had an increased probability of reporting higher life satisfaction. Although this relationship was not significant among women, the general consensus is that health insurance coverage improves access to healthcare services and utilization by lowering medical costs [72]. This does not only reduce financial burden but increases health seeking behaviours, affords medical screening

and prompt medical support in the case of poor health [73,74]. Health insurance coverage may therefore ensure that men are in good mental and physical health, leading to higher levels of life satisfaction [75–77].

Our results further revealed that household wealth is a significant determinant of life satisfaction. According to the findings, males in the richest wealth quintile and females in the fourth and richest wealth quintiles had higher levels of life satisfaction compared to those in the poorest wealth quintile. We argue that individuals from wealthy households can satisfy their physiological needs and are less concerned about financial burdens which increases their chances of achieving dreams and meeting higher-order needs and thus increasing life satisfaction in the long run [78,79]. In addition, individuals from wealthy families are happier and tend to have better social relationships, health, infrastructure, and leisure opportunities [78,79]. However, people living in poverty struggle with acquiring basic survival needs, leading to lower motivations to succeed in life [80]. Moreover, individuals from poor economic backgrounds are dissatisfied with their circumstances and faced financial difficulties such as increased debt and the inability to pay bills [81,82]. Due to these stressors, they are more likely to have low self-esteem, anxiety and frustration, and subsequently low life satisfaction [83,84].

Finally, being resident in the Greater Accra and Volta regions was associated with a higher chance of life satisfaction compared to the Western region for both men and women. Additionally, for women, being resident in Brong Ahafo, Eastern, Northern, Upper East, and Upper West regions was associated with higher life satisfaction. Although reasons for this relationship may not be fully known to the authors, a combination of factors including vast arable lands, agrarian activities, high levels of connectedness, and religiosity may be contributing to the life satisfaction of dwellers in the Brong Ahafo, Eastern, Northern, Upper East, Upper West and Volta regions [21,85–88]. The conditions may be different for the Greater Accra Region, which is predominantly urban, highly industrialized, and houses the capital city. It is a region that offers dwellers greater social, employment and economic opportunities, leading to higher levels of income and perhaps high levels of life satisfaction [89–91]. Our results inversely show that residents in the Western region were less likely to report being highly satisfied with life and this could be explained by the discovery of gas and oil in the Western belt of the country [92,93]. According to Arthur and Amo-Fosu [92] the discovery of gas and oil in the region led to higher costs of living such as increases in commodities and accommodation which has affected local residents. This has led to increases in worry among residents and may account for lower levels of satisfaction.

## Strengths and limitations

One strength of this paper is its ability to stratify established relationships along gender lines, generating more richer information about the determinants of life satisfaction between men and women in Ghana. Another strength is the study's use of a nationally representative dataset which facilitates generalization and enhances reliability by lessening the effects of potential errors induced by self-reporting. Nevertheless, these findings ought to be interpreted with caution due to limitations. First of all, the use of cross-sectional study limits the ability to assess the trends and also establish causation between the various factors and life satisfaction. It is therefore recommended that the associated factors explored in this study should be studied more longitudinally. Additionally, future Ghanaian studies should attempt using other robust analysis such as multilevel modelling as well as testing interaction effects (e.g., age-gender interaction). Secondly, we may have misspecified our model after excluding health variables (e.g., "Difficulty hearing, even if using a hearing aid") of individuals, however, it was necessary since data on these variables were collected only from participants aged 18-49years. MICS

should endeavour to include data on health variables for 15–19 years individuals in future datasets. We would also like to mention that our findings only extend previous literature on the subject matter of life satisfaction in Ghana.

## Conclusion

Our research presents findings suggesting that gender, age, parity, education, marital status, wealth index, and region of residence are determinants of life satisfaction or thriving among Ghanaians. It is also reported that this pattern of relationships slightly varied between men and women. These findings collectively provide useful information for policymakers, researchers, and practitioners. For instance, policies designed towards providing services for the Ghanaian population to improve life satisfaction should be distributed equitably and equivalently across gender, taking into consideration the intricate relationships between determinants and life satisfaction as established in this study. Evidence from this study also calls on the government of Ghana to begin tracking the life satisfaction of her citizens. In recent times, the inclusion of self-reported well-being and life satisfaction in governmental policies for tracking objective social and economic progress has been advocated [94,95]. Because of this proposal, many nations and international development organizations have taken the necessary steps to make life satisfaction central to developmental policies [96,97]. The United Nations, for instance, has included "Good health and well-being" on its list of 17 Sustainable Development Goals [98–100]. Therefore, our findings provide a step towards this realization in Ghana.

## Supporting information

**S1 Fig. Predicted probabilities of life satisfaction at highest level.** Margins plot with confidence intervals. Blue lines represent men and red lines represent women. Note: PP = Pre-primary education; predicted probabilities (on the y-axis) were derived from full sample model interacting with gender with age groups, education level, wealth quintile and marital status (each on the x-axis).
(TIF)

**S1 Table. Intercorrelation between study variables.**
(PDF)

**S2 Table. Bivariate oprobit model regressing life satisfaction on predictor variables in the full sample and gender stratified sample.**
(PDF)

**S1 File. Stata dataset of study.** Abridged version of the Stata dataset analysed for the study.
(DTA)

**S2 File. Stata do-file of study.** Stata do-file containing the commands used to run the statistical analyses.
(DO)

## Acknowledgments

We would like to thank all stakeholders involved in conducting the Multiple Indicator Cluster Survey Six. We also like to thank the MICS for granting us permission to use this data for publication.

## Author Contributions

**Conceptualization:** Kenneth Owusu Ansah, Nutifafa Eugene Yaw Dey, Pascal Agbadi.

**Data curation:** Pascal Agbadi.

**Formal analysis:** Pascal Agbadi.

**Methodology:** Nutifafa Eugene Yaw Dey.

**Project administration:** Nutifafa Eugene Yaw Dey.

**Software:** Pascal Agbadi.

**Supervision:** Nutifafa Eugene Yaw Dey, Pascal Agbadi.

**Validation:** Kenneth Owusu Ansah, Nutifafa Eugene Yaw Dey, Pascal Agbadi.

**Writing – original draft:** Kenneth Owusu Ansah, Nutifafa Eugene Yaw Dey, Abigail Esinam Adade, Pascal Agbadi.

**Writing – review & editing:** Kenneth Owusu Ansah, Nutifafa Eugene Yaw Dey, Abigail Esinam Adade, Pascal Agbadi.

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
