## [Decision Letter · Decision Letter 0]

1 Apr 2021

PONE-D-20-37979

Determinants of life satisfaction among reproductive age Ghanaians: A further analysis of the 2017/2018 Multiple Cluster Indicator Survey.

PLOS ONE

Dear Nutifafa Eugene Yaw Dey,

Thank you for submitting your manuscript to PLOS ONE. After careful consideration, we feel that it has merit but does not fully meet PLOS ONE’s publication criteria as it currently stands. Therefore, we invite you to submit a revised version of the manuscript that addresses the points raised during the review process.

The paper has without doubt the potential to be published

The conceptual objections of Reviewer 2 are quite easy to address.

Central are the methodological objections of Reviewer 1 as well as Reviewer 2:

- inclusion of happiness as a predictor:

- dichotomization of the outcome

See the detailed remarks by both reviewers.

I hope very much you will accept this invitation for a major revision. In case of a revision I will send the paoer again to both reviewers.

We look forward to receiving your revised manuscript.

Kind regards,

Gert G. Wagner, Professor

Academic Editor

PLOS ONE

2. In the methods section, please provide details regarding how the household wealth index was categorised.

Reviewers' comments:

Reviewer's Responses to Questions

**Comments to the Author**

1. Is the manuscript technically sound, and do the data support the conclusions?

Reviewer #1: Yes

Reviewer #2: Partly

2. Has the statistical analysis been performed appropriately and rigorously? 

Reviewer #1: No

Reviewer #2: No

3. Have the authors made all data underlying the findings in their manuscript fully available?

Reviewer #1: Yes

Reviewer #2: Yes

4. Is the manuscript presented in an intelligible fashion and written in standard English?

Reviewer #1: Yes

Reviewer #2: Yes

5. Review Comments to the Author

Reviewer #1: This manuscript investigates the associations between life satisfaction and various of its predictors in Ghanaians and highlights some differences between the genders. This study is overall well-conducted, and I’m a big fan of this type of more descriptive study in which associations are reported without squeezing them into any particular narrative.

I’d normally write a much longer review, but I believe that there is one central issue with the analysis that weakens all results (see below, inclusion of happiness as a predictor). Once this issue has been fixed, I’m more than happy to re-read this interesting work and provide more details. (I think this reduces the workload on both sides: I don’t have to comment on issues that will likely be fixed once analyses have been re-run; the authors don’t have to write lengthy replies to issues that no longer apply.).

Best regards,

Julia Rohrer

(I sign all my reviews)

Major issues:

- inclusion of happiness as a predictor: I believe that including happiness as a predictor will bias all your estimates. Why is that? First, think about what “the effect of X on life satisfaction, controlling for happiness” ought to mean – happiness is extremely closely related to life satisfaction. What does it mean to be more satisfied without being happier? More rigorously, I believe that a sensible model would be that both happiness and life satisfaction are measures of people’s underlying overall assessment of their lives. Controlling for happiness will remove a lot of valid variability. Thinking about it another way: how satisfied you are with your life may affect how happy you are right now. Happiness is thus an outcome of life satisfaction, including it in a model will introduce collider bias. I’ve written about collider bias elsewhere, maybe you’ll find it helpful: Rohrer, 2018, https://journals.sagepub.com/doi/full/10.1177/2515245917745629. I’d consider any of the following solutions satisfactory: omit happiness as a predictor; combine happiness and life satisfaction into a more reliable indicator of well-being (see also point below on dichotomization of the outcome); analyze both as outcomes separately (as robustness check, but also maybe to find out whether these measures react differently to various predictors).

- dichotomization of the outcome: I’m opposed to the dichotomization of the outcome; I believe that this steps discards information that could be valuable. I guess there could be some concern that respondents are not fully using the scale in a “reasonable” manner. That may as well be possible (I think in representative samples, many people are overwhelmed by too many response options), but dichotomizing isn’t really going to fix this issue—it just imposes the assumption that all 0-6 responses are the same, and that all 7-10 responses are the same, discarding valuable information (I do believe that somebody who says 0 is likely less satisfied than somebody saying 6, but your analysis would discard this information). If people don’t use a scale efficiently, they may as well “accidentally” respond 7 instead of 6, so you still have misclassification.

There is another issue with analyzing a dichotomous outcome: You know have to decide on which scale to assess interaction. Currently, you’re only looking at Odds Ratios. I believe that many economists wouldn’t be satisfied with that, as they are very much in favor of evaluating interactions on a probability scale. As it happens, I just had a paper accepted covering this topic, you may find it helpful: Rohrer & Arslan (2021), preprint: https://psyarxiv.com/7fm2j/ (relevant section is the first one on the scale dependence of interactions). If you keep the binary outcome, I’d like to see an evaluation of interaction on the scale of the probability of being thriving (Odds Ratio is rather unintuitive, its only merit are some nice statistical properties). If you instead do a regular linear regression, that won’t be an issue. (I’m aware that doing a simple linear regression with an ordinal output isn’t optimal either; but I think the proper solution would be an ordinal model rather than dichotomization. But I do think ordinal models are rather involved and often not quite interpretable, so I wouldn’t want you to run one of those either).

Minor issues:

- abstract, “in a full and gender-stratified model”: when reading this for the first time, I was really confused what “full” was referring to. A “full model”? A “fully-stratified” model? I assume you mean to say that you did analysis in two ways, one time for the full sample, one time for gender subsamples.

- line 74, “the age of Addai’s et al. dataset was within 2005-2008 while Calys-Tagoe’s et al. research focused on older adults 50years and above.”: I don’t understand what is being said here (is this a comparison of dates with ages?)

- p. 8, bivariate analysis: I don’t think it’s a good idea to use significance in a bivariate analysis as a criterion for inclusion of a variable. I even think I have seen people writing about this (it’s prone to overfitting, and in any case a bivariate association doesn’t tell you whether a variable has a causal effect or not). I still think the bivariate analysis is nice for full transparency, so I’d simply delete the sentence using it as a rationale for inclusion in the full model.

- I do like that you report gender-stratified analyses

Reviewer #2: Review of

Determinants of life satisfaction among reproductive age Ghanaians: A further analysis

of the 2017/2018 Multiple Cluster Indicator Survey

The primary aim of this study was to examine life satisfaction in a sample of Ghanaians. The authors applied logistic regression models to cross-sectional data from the Ghana Multiple Indicator Cluster Survey Six to examine these associations of life satisfaction with a number of covariates. Results are taken to indicate that less than 40 percent of participants reported being satisfied with their life and that covariates such as age, gender, and education were associated with life satisfaction. The authors pursue an interesting research question using a most likely understudied population. However, I see major conceptual and also major empirical concerns.

CONCEPTUAL

1. Most importantly, the theoretical focus of the paper needs to be clarified (p. 3-5). To illustrate, the authors state the importance of studying life satisfaction in Ghana and list a number of potential underlying factors such as poverty, but don’t elaborate on potential underlying mechanisms and how all of these constructs are linked to one another. It would be very helpful to also conceptualize these theoretical assumptions in the study or reframe the introduction so that it becomes clear how life satisfaction and the covariates are linked to one another, what the role of the specific regional context is and how this can be embedded into the existing literature.

2. The literature review is incomplete. To illustrate, a number of studies have examined life satisfaction in adults (e.g., Blanchflower & Oswald, 2008; Diener, Suh, Lucas, & Smith, 1999; Pavot & Diener, 1993). Many of these studies have also examined the role of individual difference characteristics for life satisfaction (e.g., living status). Some of those studies in a longitudinal manner. It is thus necessary to further highlight what the present study adds to the existing literature. Please clarify.

METHODOLOGICAL

p. 5-6: I stumbled across the sample selection and study description. To illustrate, it should be clearer what the recruitment strategy for the sample was. For example, are people of the same household part of the sample? If so, data wouldn’t be independent from one another and the model would need to account for potential statistical interdependence. Alternative one would need to ensure to use only one member of the household.

p. 8: The authors should provide an intercorrelation table of all variables under study. This is important because it could provide further insight into the association of their related but ideally distinct outcome measure.

p. 8: Relatedly, I was surprised to see that the authors controlled for happiness when examining life satisfaction. From a conceptual perspective, both constructs are closely related to one another and might even be used interchangeably, depending on the measure that is used. I would also expect the constructs to be highly corrected. If not, this would hint at the measures examining something different but if so, this would result in multicollinearity issues.

p. 8: My biggest concern is that the authors dichotomized the outcome variable (struggling versus thriving) and applied logistic multiple regression analyses. In general, one should always make use of all data available. Artificially dichotomizing variables will result in a loss of information that would have been available otherwise. The authors have to make use of all data available by running at least a multiple regression analysis or ideally examine all variables multivariate framework (e.g., SEM). Please clarify.

p. 8: Relatedly, did the author test for any interaction effects? To illustrate, it could be that life satisfaction varies for older men but not older women (i.e., age-gender interaction) or for lower educated individuals living alone but high educated individuals living alone. Please clarify.

p. 9: Reporting percentages is not very informative. When examining a multivariate regression using continuous outcomes and predictors, the authors would need to report the regression coefficient and also effect sizes.

p. 9: Data on the geographical regions is part of the dataset. This is very interesting and could be of potential use for the authors. However, study participants are then nested in geographical regions and thus multilevel modeling should be applied.

MINOR ISSUES

p. 1: The authors switch between the terms well-being and life satisfaction throughout the manuscript. I would suggest sticking to one and provide a definition upfront.

p. 8: Were variables of interest standardized or centered (e.g., mean-centered). Please clarify.

e.g., p. 17: Throughout the manuscript, the authors imply directionality and sometimes causality in the associations between life satisfaction and the chosen covariates. I would encourage the authors to avoid any causal or directional language since the analysis does not allow to make any conclusions about causality or directionality.

p. 4: relatedly, since the data is cross-sectional the study does not examine ‘factors which contribute to the maintenance of a high level of life satisfaction in Ghana’ but ‘factors which are associated with life satisfaction in Ghana’.

Title, Abstract, p. 1-21: I wonder if it is necessary to add the term reproductive to the sample description when not discussing it further. E.g., would the authors expect this result to be an effect of reproductivity? Also, can one compare the reproductive age of men and women?

6. PLOS authors have the option to publish the peer review history of their article (what does this mean?). If published, this will include your full peer review and any attached files.

Reviewer #1: **Yes: **Julia M. Rohrer

Reviewer #2: No

---

## [Author Response · Author response to Decision Letter 0]

27 May 2021

Response to Reviewers’ Comments

Manuscript Number: PONE-D-20-37979

Dear Editor,

Thank you for your letter and the opportunity to revise our paper on “Determinants of life satisfaction among Ghanaians aged 15 to 49 years: Further analysis of the 2017/2018 Multiple Cluster Indicator Survey”. We are grateful for the reviews provided by the reviewers.

Thank you for your continued interest in our research and we hope that this improved manuscript is accepted for publication in PLOS ONE. Below are our responses to the comments and concerns.  

Editor comments

Response

Thank you for the reminder. We have followed the formatting requirements accordingly.

2. In the methods section, please provide details regarding how the household wealth index was categorized

Response

The authors have provided these details. “Household wealth was categorized into poorest (0), second quintile (1), middle (2), fourth quintile (3), and richest (4).”

Reviewer comments:    

Reviewer #1:

1. inclusion of happiness as a predictor: I believe that including happiness as a predictor will bias all your estimates. Why is that? First, think about what “the effect of X on life satisfaction, controlling for happiness” ought to mean – happiness is extremely closely related to life satisfaction. What does it mean to be more satisfied without being happier? More rigorously, I believe that a sensible model would be that both happiness and life satisfaction are measures of people’s underlying overall assessment of their lives. Controlling for happiness will remove a lot of valid variability. Thinking about it another way: how satisfied you are with your life may affect how happy you are right now. Happiness is thus an outcome of life satisfaction, including it in a model will introduce collider bias. I’ve written about collider bias elsewhere, maybe you’ll find it helpful: Rohrer, 2018, https://journals.sagepub.com/doi/full/10.1177/2515245917745629. I’d consider any of the following solutions satisfactory: omit happiness as a predictor; combine happiness and life satisfaction into a more reliable indicator of well-being (see also point below on dichotomization of the outcome); analyze both as outcomes separately (as robustness check, but also maybe to find out whether these measures react differently to various predictors).

Response

Thank you for noting these errors. As suggested, we have omitted happiness as a predictor. 

2. Dichotomization of the outcome: I’m opposed to the dichotomization of the outcome; I believe that this steps discards information that could be valuable. I guess there could be some concern that respondents are not fully using the scale in a “reasonable” manner. That may as well be possible (I think in representative samples, many people are overwhelmed by too many response options), but dichotomizing isn’t really going to fix this issue—it just imposes the assumption that all 0-6 responses are the same, and that all 7-10 responses are the same, discarding valuable information (I do believe that somebody who says 0 is likely less satisfied than somebody saying 6, but your analysis would discard this information). If people don’t use a scale efficiently, they may as well “accidentally” respond 7 instead of 6, so you still have misclassification.

There is another issue with analyzing a dichotomous outcome: You know have to decide on which scale to assess interaction. Currently, you’re only looking at Odds Ratios. I believe that many economists wouldn’t be satisfied with that, as they are very much in favor of evaluating interactions on a probability scale. As it happens, I just had a paper accepted covering this topic, you may find it helpful: Rohrer & Arslan (2021), preprint: https://psyarxiv.com/7fm2j/ (relevant section is the first one on the scale dependence of interactions). If you keep the binary outcome, I’d like to see an evaluation of interaction on the scale of the probability of being thriving (Odds Ratio is rather unintuitive, its only merit are some nice statistical properties). If you instead do a regular linear regression, that won’t be an issue. (I’m aware that doing a simple linear regression with an ordinal output isn’t optimal either; but I think the proper solution would be an ordinal model rather than dichotomization. But I do think ordinal models are rather involved and often not quite interpretable, so I wouldn’t want you to run one of those either).

Response

Thank you for this suggestion. We have removed dichotomization of the outcome variable. Currently, we are using the variable in its original state, using all the points on the Cantril’s Self-Anchoring Ladder of Life Satisfaction scale. Correspondingly, the data analysis was changed from binary logistic regression to ordered probit regression as you suggested. 

3. abstract, “in a full and gender-stratified model”: when reading this for the first time, I was really confused what “full” was referring to. A “full model”? A “fully-stratified” model? I assume you mean to say that you did analysis in two ways, one time for the full sample, one time for gender subsamples 

Response

This correction has been done. Kindly find this correction on the abstract page. “This study, therefore, extends previous literature by examining the determinants of life satisfaction among Ghanaians in two ways: a full sample and gender-stratified sample.”

4. line 74, “the age of Addai’s et al. dataset was within 2005-2008 while Calys-Tagoe’s et al. research focused on older adults 50years and above.”: I don’t understand what is being said here (is this a comparison of dates with ages?)

Response

This correction has been done. Kindly find this correction on page 4. “The age of Addai et al’s (41) dataset is now quite old, using data that was collected within 2005-2008. Also, Calys-Tagoe et al’ study (12) only focused on older adults (50 years and above).”

5. p. 8, bivariate analysis: I don’t think it’s a good idea to use significance in a bivariate analysis as a criterion for inclusion of a variable. I even think I have seen people writing about this (it’s prone to overfitting, and in any case a bivariate association doesn’t tell you whether a variable has a causal effect or not). I still think the bivariate analysis is nice for full transparency, so I’d simply delete the sentence using it as a rationale for inclusion in the full model.

Response

Thank you for this suggestion. We have removed this sentence from the manuscript. 

6. - I do like that you report gender-stratified analyses

Response 

Thank you for the feedback.

Reviewer #2:

1. Most importantly, the theoretical focus of the paper needs to be clarified (p. 3-5). To illustrate, the authors state the importance of studying life satisfaction in Ghana and list a number of potential underlying factors such as poverty, but don’t elaborate on potential underlying mechanisms and how all of these constructs are linked to one another. It would be very helpful to also conceptualize these theoretical assumptions in the study or reframe the introduction so that it becomes clear how life satisfaction and the covariates are linked to one another, what the role of the specific regional context is and how this can be embedded into the existing literature

Response

Thank you for this suggestion. We have accordingly reframed the introduction by clearly indicate how life satisfaction and the covariates are linked to one another. Here is an example of text elaborating this on page 3. “In Diego-Rosell, Tortora, and Bird’s (35) study, for instance, it was found out that families with a high household income had better life satisfaction across the 153 countries. Luhmann et al. (36) also found from analysing longitudinal data from three nationally representative panel studies that life satisfaction levels increased with marriage and childbirth but reduced with marital separation, job loss, starting a new job, and relocating”. 

2. The literature review is incomplete. To illustrate, a number of studies have examined life satisfaction in adults (e.g., Blanchflower & Oswald, 2008; Diener, Suh, Lucas, & Smith, 1999; Pavot & Diener, 1993). Many of these studies have also examined the role of individual difference characteristics for life satisfaction (e.g., living status). Some of those studies in a longitudinal manner. It is thus necessary to further highlight what the present study adds to the existing literature. Please clarify.

Response

Thank you for bringing this to our attention. We have further highlighted on how this present study adds to existing literature. See page 5, “Our study goes a step further by examining closely these factors from a gendered perspective; an examination that is virtually nonexistent in Ghana. Due to the variations in social norms and biological characteristics, a gendered perspective has been recommended to provide more nuanced information into the associated factors of both men and women’s life satisfaction (36–38).”

3. p. 5-6: I stumbled across the sample selection and study description. To illustrate, it should be clearer what the recruitment strategy for the sample was. For example, are people of the same household part of the sample? If so, data wouldn’t be independent from one another and the model would need to account for potential statistical interdependence. Alternative one would need to ensure to use only one member of the household.

Response

Thank you for this suggestion. We have clearly indicated the recruitment strategy for the sample. A complex survey design was used to account for statistical interdependence. See page 9, “In each household, people from the same household were selected.” and, page 6, “We accounted for the complex sampling design embedded in the dataset to monitor possible analytical errors and make proper inferences (45). This was achieved by correcting for clusters, stratification, and sample weights using the complex survey mode command 'svyset'.”

4. p. 8: The authors should provide an intercorrelation table of all variables under study. This is important because it could provide further insight into the association of their related but ideally distinct outcome measure.

Response

Thank you for bringing this to our attention. An intercorrelation table has been included as supplementary material. For easy identification of the intercorrelation table, a statement has been provided. See page 12, “Spearman’s rho was used as a preliminary test to assess the intercorrelations between all study variables (see S1 Table).”

5. p. 8: Relatedly, I was surprised to see that the authors controlled for happiness when examining life satisfaction. From a conceptual perspective, both constructs are closely related to one another and might even be used interchangeably, depending on the measure that is used. I would also expect the constructs to be highly corrected. If not, this would hint at the measures examining something different but if so, this would result in multicollinearity issues 

Response

Thank you for noting this. As suggested, we have omitted happiness as a predictor.

6. p. 8: My biggest concern is that the authors dichotomized the outcome variable (struggling versus thriving) and applied logistic multiple regression analyses. In general, one should always make use of all data available. Artificially dichotomizing variables will result in a loss of information that would have been available otherwise. The authors have to make use of all data available by running at least a multiple regression analysis or ideally examine all variables multivariate framework (e.g., SEM). Please clarify.

Response

Thank you for this suggestion. We have removed dichotomization of the outcome variable. Currently, we are using the variable in its original state, using all the points on the Cantril’s Self-Anchoring Ladder of Life Satisfaction scale. Correspondingly, the data analysis was changed from binary logistic regression to ordered probit regression as you suggested. 

7. p. 8: Relatedly, did the author test for any interaction effects? To illustrate, it could be that life satisfaction varies for older men but not older women (i.e., age-gender interaction) or for lower educated individuals living alone but high educated individuals living alone. Please clarify.

Response

There was no interaction effect tested in the new analysis. Although, the authors acknowledge the impact of such interaction, testing interaction effect was beyond the scope of the study’s objectives. Your suggestion has been recommended for future researchers. See page 21, “Additionally, future Ghanaian studies should attempt using other robust analysis such as multilevel modelling as well as testing interaction effects (e.g., age-gender interaction).”

8. p. 9: Reporting percentages is not very informative. When examining a multivariate regression using continuous outcomes and predictors, the authors would need to report the regression coefficient and also effect sizes.

Response

We have reported the regression coefficient and average marginal effect. These can be found in Table 2 and 3. 

9. p. 9: Data on the geographical regions is part of the dataset. This is very interesting and could be of potential use for the authors. However, study participants are then nested in geographical regions and thus multilevel modeling should be applied.

Response

Thank you for this suggestion. Although we agree that multilevel modeling is one of the alternative methods, complex survey analysis also accounts for these inherent variations and clusters. We, therefore, have recommended the use of multilevel modelling for future studies in Ghana. See page 21, “Additionally, future Ghanaian studies should attempt using other robust analysis such as multilevel modelling as well as testing interaction effects (e.g., age-gender interaction).”

10. p. 1: The authors switch between the terms well-being and life satisfaction throughout the manuscript. I would suggest sticking to one and provide a definition upfront.

Response

We have accordingly sticked to the use of only life satisfaction in this manuscript. You can find these throughout the manuscript. 

11. p. 8: Were variables of interest standardized or centered (e.g., mean-centered). Please clarify.

Response

The analysis was redone using ordered probit regression. Because ordered probit regression accounts for all the points in the outcome variable, there was no need for us to standardize or center the variable. 

12. e.g., p. 17: Throughout the manuscript, the authors imply directionality and sometimes causality in the associations between life satisfaction and the chosen covariates. I would encourage the authors to avoid any causal or directional language since the analysis does not allow to make any conclusions about causality or directionality.

Response

Thank you for noting this. We have accordingly removed any causal language. Here is an example of text highlighting this change, on page 16, “The association between marital status, parity, household wealth index, region of residence, and life satisfaction were different for females and males.”

13. p. 4: relatedly, since the data is cross-sectional the study does not examine ‘factors which contribute to the maintenance of a high level of life satisfaction in Ghana’ but ‘factors which are associated with life satisfaction in Ghana’.

Response

Thank you for suggesting this. We have reframed our introduction and thus, this statement has been omitted from the work. 

14. Title, Abstract, p. 1-21: I wonder if it is necessary to add the term reproductive to the sample description when not discussing it further. E.g., would the authors expect this result to be an effect of reproductivity? Also, can one compare the reproductive age of men and women?

Response

This correction has been made. Kindly find this correction on the abstract page “This study, therefore, extends previous literature by examining the determinants of life satisfaction among Ghanaians in two ways: a full model and gender-stratified model” and the title page, “Determinants of life satisfaction among Ghanaians aged 15 to 49 years: A further analysis of the 2017/2018 Multiple Cluster Indicator Survey.”

---

## [Decision Letter · Decision Letter 1]

28 Jul 2021

PONE-D-20-37979R1

Determinants of life satisfaction among Ghanaians aged 15 to 49 years: A further analysis of the 2017/2018 Multiple Cluster Indicator Survey

PLOS ONE

Dear Dr. Dey,

Thank you for submitting your manuscript to PLOS ONE. After careful consideration, we feel that it has merit but does not fully meet PLOS ONE’s publication criteria as it currently stands. Therefore, we invite you to submit a revised version of the manuscript that addresses the points raised during the revision process.It is felt that the manuscript improved enormously from the previous drafts. Now the analysis is sound with the removal of happiness and the ordered probit.Reviewer 1 has minor suggestions to edit. There is sometimes a choice of innacurate words like rudimentary for elemmentary.The paper is missing a methods section describing the ordered probit model used explaining its strenghts and limitations in this context. Please submit your revised manuscript by Sep 11 2021 11:59PM. If you will need more time than this to complete your revisions, please reply to this message or contact the journal office at plosone@plos.org. Please include the following items when submitting your revised manuscript:

We look forward to receiving your revised manuscript.

Kind regards,

José Antonio Ortega, Ph.D.

Academic Editor

PLOS ONE

Journal Requirements:

Reviewers' comments:

Reviewer's Responses to Questions

**Comments to the Author**

1. If the authors have adequately addressed your comments raised in a previous round of review and you feel that this manuscript is now acceptable for publication, you may indicate that here to bypass the “Comments to the Author” section, enter your conflict of interest statement in the “Confidential to Editor” section, and submit your "Accept" recommendation.

Reviewer #1: All comments have been addressed

2. Is the manuscript technically sound, and do the data support the conclusions?

Reviewer #1: Yes

3. Has the statistical analysis been performed appropriately and rigorously? 

Reviewer #1: Yes

4. Have the authors made all data underlying the findings in their manuscript fully available?

Reviewer #1: No

5. Is the manuscript presented in an intelligible fashion and written in standard English?

Reviewer #1: Yes

6. Review Comments to the Author

Reviewer #1: I’d like the authors for carefully addressing all of my comments. My previous concerns no longer apply to the revised version of the manuscript, which I believe will make a great contribution to the literature on well-being. I have only very few minor remarks left, such as typos or small tweaks to the language. Apart from that, I’d like to encourage the others to make their Stata do-files available for the scientific community, for example, on the Open Science Framework (osf.io). That way, other researchers can apply for access to the data used and then use the do-files to reproduce results; but it is also really helpful to double-check how exactly models were specified.

(I believe that this would also be aligned with PLOS ONE’s Data Availability Policy)

Best regards,

Julia Rohrer (I sign all my reviews)

p. 3, “life satisfaction across the 153 countries”: I think this should either be “across 153 countries” or “across the 153 countries included in their study.”

p. 4, “Considering these limitations and the pressing need to produce more recent evidence regardless of the persisting contextual problems ranging from limited access to drinking water [26], unemployment particularly among the youth [27], limited access to health care, poverty [28], high prevalence of chronic diseases [29–33] and poor quality education [34,35] that may threaten one’s life satisfaction, our study used the 2017/2018 Multiple Cluster Indicator Survey to examine the factors associated with life satisfaction in Ghana.”: I find the usage of “regardless of” here confusing. I wouldn’t say we need more evidence regardless of the problems, I would say we need more evidence in particular because of these problems?

p. 6, “The final samples were 660 clusters and 13202 households to the sampling strata.”: I’m confused by “to the sampling strata”; do you mean “across all sampling strata”?

p. 7, “This recategorized variable was kept for only descriptive purposes; therefore, the original variable (i.e., the 0 to 10 ordinal variable) was used in the main study analyses.”: I think this sentence actually becomes clearer if you just omit the “therefore”

p. 15, Table 3, 2 column from left, Parity Effects for Men at middle life satisfaction: there’s five cells with the same value (.195), I suspect this may be a copy-paste-error

7. PLOS authors have the option to publish the peer review history of their article (what does this mean?). If published, this will include your full peer review and any attached files.

Reviewer #1: **Yes: **Julia M. Rohrer

---

## [Author Response · Author response to Decision Letter 1]

8 Aug 2021

Dear Dr. José Antonio Ortega,

Thank you very much for giving us another opportunity to improve our manuscript. Below are the responses to the editorial and reviewer’s comments. 

Editor’s comment

There is sometimes a choice of innacurate words like rudimentary for elemmentary.

Authors’ response

We have rectified this error. The new sentence reads like this: “Briefly, most of the selected variables were measured in a simple manner…”

Editor’s comment

The paper is missing a methods section describing the ordered probit model used explaining its strengths and limitations in this context. 

Authors’ response

Thank you for noting this omission. We have updated the manuscript accordingly. Find this on page 8: “The ordered probit model is typically used to examine the variation in the data points of an ordinal categorical dependent variable (44). Though argued to produce parameter estimates difficult to interpret, oprobit was fitted mainly for its ability to preserve the ordering of the response options in the outcome variable as a function of the predictor variables (45).”

Journal Requirements:

Authors’ response.

The reference list has been double checked and we found no retracted papers. However, three more references were included in the reference list.

Reviewers' comment

Reviewer #1: I’d like the authors for carefully addressing all of my comments. My previous concerns no longer apply to the revised version of the manuscript, which I believe will make a great contribution to the literature on well-being. I have only very few minor remarks left, such as typos or small tweaks to the language. Apart from that, I’d like to encourage the others to make their Stata do-files available for the scientific community, for example, on the Open Science Framework (osf.io). That way, other researchers can apply for access to the data used and then use the do-files to reproduce results; but it is also really helpful to double-check how exactly models were specified.

(I believe that this would also be aligned with PLOS ONE’s Data Availability Policy)

Authors’ response

Thank you for the feedback and yes, the Stata do-file as well as the dataset are attached to the revised manuscript.

Reviewer’s comment

p. 3, “life satisfaction across the 153 countries”: I think this should either be “across 153 countries” or “across the 153 countries included in their study.”

Authors’ response

Thank you for pointing this out. We have accordingly corrected this sentence: “…it was found that families with a high household income had better life satisfaction across 153 countries.”

Reviewer’s comment

p. 4, “Considering these limitations and the pressing need to produce more recent evidence regardless of the persisting contextual problems ranging from limited access to drinking water [26], unemployment particularly among the youth [27], limited access to health care, poverty [28], high prevalence of chronic diseases [29–33] and poor quality education [34,35] that may threaten one’s life satisfaction, our study used the 2017/2018 Multiple Cluster Indicator Survey to examine the factors associated with life satisfaction in Ghana.”: I find the usage of “regardless of” here confusing. I wouldn’t say we need more evidence regardless of the problems, I would say we need more evidence in particular because of these problems?

Authors response

Thank you for noting this error and the suggestion. The sentence has been restructured as such: “Considering these limitations and the pressing need to produce more recent evidence in particular because of persisting contextual problems ranging…”

Reviewer’s comment

p. 6, “The final samples were 660 clusters and 13202 households to the sampling strata.”: I’m confused by “to the sampling strata”; do you mean “across all sampling strata”?

Authors’ response

Indeed, we meant “across all sampling strata” and this correction has been made. Thank you.

Reviewer’s comment

p. 7, “This recategorized variable was kept for only descriptive purposes; therefore, the original variable (i.e., the 0 to 10 ordinal variable) was used in the main study analyses.”: I think this sentence actually becomes clearer if you just omit the “therefore”

Authors’ response

This error has been corrected thanks to your suggestion. Here is the updated sentence: “This recategorized variable was kept solely for the purposes of description; the original variable…”

Reviewer’s comment

p. 15, Table 3, 2 column from left, Parity Effects for Men at middle life satisfaction: there’s five cells with the same value (.195), I suspect this may be a copy-paste-error

Authors’ response

Interestingly, this is not a copy-paste error. Similar values were generated through the margins command and rounded to .195. We are assuming they are the same because the probabilities are being predicted at the middle level of life satisfaction.

---

## [Editor Report · Decision Letter 2]

29 Nov 2021

Determinants of life satisfaction among Ghanaians aged 15 to 49 years: A further analysis of the 2017/2018 Multiple Cluster Indicator Survey

PONE-D-20-37979R2

Dear Dr. Dey,

We’re pleased to inform you that your manuscript has been judged scientifically suitable for publication and will be formally accepted for publication once it meets all outstanding technical requirements.

Kind regards,

José Antonio Ortega, Ph.D.

Academic Editor

PLOS ONE

Additional Editor Comments (optional):

It is deemed that the issues have been satisfactorily addressed and the article is ready for publication. Congratulations.

While it is not required, if you want you can add some paragraph on the difference of the age pattern found with other studies. The minimum satisfaction is generally at 45-55 years. Ghana seems different. (Eg: https://ideas.repec.org/p/pra/mprapa/7302.html but see also https://doi.org/10.1007/s00148-020-00797-z with cross-cultural evidence).
---

## [Editor Report · Acceptance letter]

21 Dec 2021

PONE-D-20-37979R2 

Determinants of life satisfaction among Ghanaians aged 15 to 49 years: A further analysis of the 2017/2018 Multiple Cluster Indicator Survey. 

Dear Dr. Dey:

I'm pleased to inform you that your manuscript has been deemed suitable for publication in PLOS ONE. Congratulations! Your manuscript is now with our production department. 

Kind regards, 

on behalf of

Dr. José Antonio Ortega 

Academic Editor

PLOS ONE